# A Large-Scale Three-Dimensional Apparatus to Study Failure Mechanisms of Rockfalls in Underground Engineering Contexts

**DOI:** 10.3390/s24072068

**Published:** 2024-03-24

**Authors:** Gongfeng Xin, Guangyu Yang, Fan Li, Hongliang Liu

**Affiliations:** 1Shandong Key Laboratory of Highway Technology and Safety Assessment, Jinan 250101, China; xingongfeng@sdhsg.com (G.X.); lifanll@mail.sdu.edu.cn (F.L.); 2Innovation Research Institute, Shandong Hi-Speed Group Co., Ltd., Jinan 250101, China; 3School of Qilu Transportat, Shandong University, Jinan 250002, China; 201814649@mail.sdu.edu.cn; 4Geotechnical and Structural Engineering Research Center, Shandong University, Jinan 250061, China

**Keywords:** underground engineering, rockfalls, physical simulation, large-scale experiment

## Abstract

Rockfalls are an important factor affecting underground engineering safety. However, there has been limited progress in understanding and predicting these disasters in the past few years. Therefore, a large-scale three-dimensional experimental simulation apparatus to study failure mechanisms of rockfalls occurring during underground engineering was developed. This apparatus, measuring 4 m × 4 m × 3.3 m in size, can achieve vertical and horizontal symmetric loading. It not only simulates the structure and stress environment of a rock mass but also simulates the stepwise excavation processes involved in underground engineering. A complete simulation experiment of rockfalls in an underground engineering context was performed using this apparatus. Dynamic evolution characteristics of block displacement, temperature, natural vibration frequency, and acoustic emissions occurring during rockfalls were studied during the simulation. These data indicate there are several indicators that could be used to predict rockfalls in underground engineering contexts, leading to better prevention and control.

## 1. Introduction

A natural rock mass always splits into different types of blocks of varying sizes along numerous discontinuities [1,2,3,4]. When the blocks satisfy certain conditions, they are apt to fall and incur instability [5,6]. Rockfall has long been one of the most common disasters in underground engineering. Due to the strong suddenness of the rockfall process, it is difficult to effectively capture the precursors of catastrophic displacement. Traditional displacement monitoring methods based on plasticity theory are no longer applicable, and physical information monitoring has not fully considered the effectiveness of monitoring methods and the scientific monitoring of indicators, and it is impossible to fundamentally realize the monitoring and early warning of rockfall disasters [7,8].

A large number of scholars have conducted in-depth research on rockfall mechanisms and prevention and control problems with different means, such as theoretical analysis, numerical simulation, physical simulation tests, etc. [9,10,11,12]. A large number of experimental studies have shown that the model test can better simulate the physical and mechanical properties of the rock mass and the structural conditions such as joints and fissures and consider the joint action between the surrounding rock and the support structure, which is conducive to promoting the research on the rockfall mechanism in underground engineering [13,14,15]. However, the traditional physical simulation test equipment for underground engineering can no longer meet modern research needs in terms of loading and excavation.

On the basis of traditional physical simulation test equipment, scholars have improved and developed a large number of high-precision, multi-functional physical simulation test instruments for single- or even multiple-test requirements, which have greatly improved the effect of physical simulation experiments. Wang [16] developed a spatial sliding-block apparatus. The apparatus consisted of a 1200 mm × 800 mm basal plate, an 800 mm × 600 mm inclined table, and a rotating table with a diameter of 500 mm. It simulated the occurrence of rockfalls in a fractured rock mass, reflecting the real situation of block sliding with different dip directions and dip angles. Xu [17] developed a physical experimental device for tunnel rockfall simulations, in which the blocks were simulated by aluminum pieces, and the tunnel was simulated by a wax door. They derived the kinematic behavior of the overlying rock stratum and the change laws of the loose zone for different bedding dips. Yang [18] investigated the effect of alternating soft and hard strata on the stability of rock surrounding tunnels, and physical experiments and numerical simulations were performed to simulate tunnel excavation in slanted upper-soft and lower-hard strata. The evolution laws and distribution features of stress, displacement, and failure were analyzed. Zhu [19] carried out the study to describe physical model testing and real-time observations on the evolution process of v-shaped brittle fracture, failure characteristics, onset conditions, and influencing factors in hard rock tunnels. Lu [20] developed an indoor large-scale three-dimensional physical similarity model, and a novel method was used to simulate the mining process of coal seams. Zhang [21] developed a hydro-mechanical coupled geo-mechanical model test system, and it is mainly composed of a high-pressure sealed model test chamber, a built-in high-hydraulic servo loading system, a high-seepage pressure omnidirectional loading system, an inclined geologic structure fabrication system, and a high-precision integrated test system. The internal field of the model test apparatus is 1.0 m in length, 1.0 m in width, and 1.0 m in height. Such devices provide a foundation for the development of rockfall simulation experiments. However, these devices suffer from several shortcomings: (1) the structure and stress of rock mass cannot be simulated accurately; (2) the process of excavation cannot be simulated in detail; and (3) limited information is obtained during the experiment. Thus, the experimental process does not fully reflect all failure stages.

In this paper, a large-scale three-dimensional experimental apparatus for rockfall simulation was developed. The experimental apparatus can simulate the entire process of an underground rockfall related to underground excavation—including gestation, development, and formation stages—to facilitate the study of the instability mechanism of rockfall disasters. It has the following characteristics: (1) a prefabricated sub-system that adopts a box structure for reconstructing, transporting, and loading of the simulated blocky rock mass; (2) a positioning sub-system that adjusts the prefabricated sub-system into the designated loading position; (3) a high-rigidity hydraulic sub-system to apply single and multi-directional loads on the specimen to simulate the stress environment of a blocky rock mass; and (4) an excavation process that Is simulated by the stepwise release of 22 hydraulic cylinders and triggers failure. Therefore, it can be used to study the accumulation mechanism, as well as the dynamic evolution characteristics of rockfalls in underground engineering contexts. Moreover, a complete simulation experiment of an underground rockfall was carried out. Dynamic evolution characteristics of displacement, temperature, natural vibration frequency, and acoustic emissions during the rockfalls were studied, providing support for the prevention and control of rockfalls in underground engineering contexts.

## 2. Methods

### 2.1. Experimental Simulation Apparatus

The design of our experimental apparatus considers the characteristics of gestation, development, and formation stages of an underground rockfall. In the gestation stage, potentially unstable blocks, which are enclosed by natural joints and mining-induced fractures, are apparent around excavations. The key to the simulation of the rockfall process is to reconstruct the structure and the stress environment of the blocky rock mass. Therefore, the design of the apparatus needs to provide a convenient prefabrication method and a stable and controllable stress environment for the blocky rock mass. In the development stage, potentially unstable blocks are gradually exposed during excavation and become more unstable as an outcome of mining. The apparatus needs to simulate the excavation process and ensure that the blocks are not affected by other disturbance factors. In the formation stage of the underground rockfall, the potentially unstable blocks fall, as instability occurs, which can develop within a very short time period. It is necessary to ensure high-frequency real-time monitoring and the acquisition of experimental data in each stage of the experiment to characterize the dynamic evolution of the rockfall. Therefore, the underground rockfall simulation experimental apparatus (Figure 1) consists of three main parts: the prefabricated sub-system, the positioning sub-system, and the hydraulic sub-system.

#### 2.1.1. Prefabricated Sub-System

The prefabricated sub-system (Figure 2) is a key component of the underground rockfall simulation experimental apparatus. It is used to provide a less turbulent environment for prefabricating, transporting, and loading of the specimen. To provide a framework for reconstructing the structure of blocky rock mass, a box structure with custom panels on the inside was adopted. Four sides of this box structure could be used as plates to apply loadings on the specimen, while the base consisted of 22 loading plates to simulate the excavation process in a step-by-step manner. Moreover, numerous connectors were used. For example, the top loading plate and the lateral loading plates were connected by four L-type connectors, while the lateral loading plates and basal loading plates were connected by two U-type connectors and the stability of the whole structure was improved. When these connectors are removed, the loading plates can move freely in the normal direction. During the test, only the connecting bolts of the loading plate and the front and rear cover plates need to be removed, which can be directly loaded by the hydraulic system and can effectively prevent the disturbance caused during the demolding process of the prefabricated sample.

#### 2.1.2. Positioning Sub-System

The positioning sub-system is used for the adjustment of the specimen within the underground rockfall simulation experimental apparatus. It consists of three main parts (Figure 3): a positioning hook, a balance bracket, and a platform with propulsion. The process of positioning involves three processes, referred to as I–III. In process I, the prefabricated sub-system is transferred to the apparatus and placed on the platform in front of the balance bracket. In process II, the platform is moved back and forth under propulsion to ensure that the prefabricated sub-system is in full contact with the balance bracket and the positioning hook is accurately attached to the prefabricated sub-system. In process III, the platform travels its maximum moving range, causing the prefabricated sub-system to be pushed into the loading position, ensuring the 22 basal loading plates overlap with the 22 hydraulic loading plates in the normal direction. 

#### 2.1.3. Hydraulic Sub-System

The hydraulic sub-system is the main component of the apparatus; it is used to provide the stress environment and dynamic path of the experiment. It consists of four parts (Figure 4): a control platform, a main box, a hydraulic press, and multiple hydraulic cylinders. On the base of the hydraulic sub-system, 22 cylinders have separate control modes, which implement the vertical constraints, with a maximum moving range of 400 mm. On the top and sides of the hydraulic sub-system, multiple hydraulic cylinders have synchronous loading modes, which implement both vertical and horizontal symmetric loadings, as shown in Figure 4. The vertical loading pressure is 2600 kN, with a maximum moving range of 200 mm, while the horizontal loading pressure is 1400 kN, with a maximum moving range of 200 mm. The use of multiple sets of small-flow high-pressure pumps can achieve precise control of the force and loading speed, with a control accuracy of ±6%. This allows long-term loading (≥720 h) under a constant holding pressure, and the pressure can be automatically compensated during the pressure holding process and the minimum compensation value can be freely set according to the test requirements and loading accuracy.

#### 2.1.4. Technical Advantages and Major Parameters

The main parameters of the experimental apparatus are given in Table 1. The experimental apparatus can simulate the entire process of an underground rockfall related to underground excavation—including gestation, development, and formation stages—to facilitate the study of the instability mechanism of rockfall disasters. It has the following characteristics: (1) a prefabricated sub-system that adopts a box structure for reconstructing, transporting, and loading of the simulated blocky rock mass; (2) a positioning sub-system that adjusts the prefabricated sub-system into the designated loading position; (3) a high-rigidity hydraulic sub-system to apply single- and multi-directional loads on the specimen to simulate the stress environment of a blocky rock mass; and (4) an excavation process that is simulated by the stepwise release of 22 hydraulic cylinders and triggers failure.

### 2.2. Experimental Materials and Specimen

To test the function of the experimental apparatus, a rockfall simulation experiment was carried out. The dynamic evolution characteristics of the various parameters measured during the experiment are discussed below.

A cement mortar with a ratio of 1:1:0.6 of cement, river sand, and water was used as the simulation material for the blocky rock mass. The structure and size of the specimen are shown in Figure 5. Prefabrication involves four processes. Firstly, some custom panels were used to enclose the model specimen in the box. Next, block 4 and block 3 were simultaneously prefabricated. Then, block 2 was prefabricated. Finally, block 1 was prefabricated. Moreover, the custom panels were gradually replaced by soft plastic during prefabrication. Each process required 20 h to implement. Meanwhile, the specimen was cured for more than 28 d, before loading it for experimentation.

### 2.3. Loading and Excavation Scheme

To simulate the real stress state of the surrounding rock mass, a load of 0.3 MPa was applied at the top and lateral sides of the physical model to simulate gravity and horizontal tectonic stress, respectively. All 22 basal hydraulic cylinders traveled to their maximum moving range of 400 mm to apply the vertical constraints. These 22 basal hydraulic cylinders were released by −10 mm in a stepwise process to simulate the excavation process, as shown in Figure 6. 

The steps of loading the test body under confining pressure are as follows: ① Hoist the blocky rock mass and the prefabricated frame onto the positioning sub-system to ensure that the bottom of the prefabricated frame is close to the anti-fall fixing frame, and then realize the horizontal positioning adjustment of the test body; ② start the sliding plate pushing device to accurately deliver the test frame to the test position; ③ start No.1 and No.22 oil cylinders to make them rise synchronously, so as to lift the test frame; ④ start the oil cylinder of the upper loading plate, so that the upper loading plate descends and stops after making contact with the upper edge of the prefabricated frame; ⑤ start No.2–No.21 oil cylinders of the lower loading plate, and stop after reaching the same height as No.1 and No.22 oil cylinders, so as to realize the up-and-down clamping of the prefabricated frame; ⑥ drive the left and right loading plates to contact the left and right edges of the test frame and then stop, so as to realize the left and right clamping of the test frame, and then solve the fixing problem of the prefabricated frame; ⑦ remove the prefabricated frame connecting square pipe, connecting channel steel, and loading plate connecting bolts in turn; ⑧ start the upper loading plate, load it to 0.3 Mpa, stop loading it, start the left and right loading plates, and stop loading it to 0.3 Mpa, thereby completing the loading operation of the test body.

During the experiment, the excavation simulation is divided into two directions: ① left to right and ② right to left.

### 2.4. Experiment Monitoring Scheme

The data monitoring and acquisition system is shown in Figure 7. In this test, a Z+F 5010C laser scanner is used to monitor the displacement field of the whole blocky rock mass in real time, and the displacement response law of rock mass during tunnel construction is revealed through point cloud data mining. The FLIRA615 Infrared thermal imager is used to monitor the temperature response of unstable blocks and adjacent blocks during the collapse of dangerous rocks and reveal the response law of the rock mass temperature field during tunnel construction. The PCI-EXPRESS-8 acoustic emission system is used to monitor the internal damage of rock mass during rockfall and reveal the acoustic emission response law of rock mass during tunnel construction. The instability process of the block will inevitably accompany the change in its own vibration frequency, so the DH59322N dynamic acquisition instrument with the 1A314E three-way acceleration sensor is used to monitor the vibration frequency of four blocks during the test, which is used to systematically reveal the natural vibration frequency response law of rock mass during the instability process of dangerous rock.

## 3. Results

### 3.1. Results of Displacement Information

The overall displacement of the test body is scanned and monitored by a 3D laser scanner, and the 3D point cloud data are subsequently processed to obtain the displacement change in the overall structure by comparing the point cloud data before and after the excavation steps.

The actual excavation process for direction ① is shown in Figure 8. In the 11th step, 21st step, and 22nd step of excavation, block 4, block 2, and block 1 were completely exposed and fell, respectively. Block 3 also fell during the 22nd step of excavation, because all vertical constraints were released once block 1 fell. After the 18th step, the upper left corner of block 2 made contact with block 4; this contact area increased as excavation progressed. No other displacement of blocks occurred during the simulation of the excavation process.

The actual excavation process for direction ② is shown in Figure 9. In the 12th step, rock block 2 was completely exposed, but due to the friction of the structural surface, no significant displacement occurred. In the 18th step, rock block 4 was dumped and deformed, the right end of the adjacent rock block settled, and the structural surface was slightly separated. In the 19th step, the dumping deformation was further aggravated. At the same time, the frictional resistance of the structural surface between rock block 4 and rock block 2 was reduced, rock block 2 slipped, and there was a violent shaking sound. In the 20th step, rock block 2 fell off, and the whole rock block was obviously separated.

This suggests that the deformation of the blocky rock mass under low confining pressure occurs via the rigid displacement of blocks. Block fall occurs, once the block is completely exposed, although there is almost no previous displacement. Therefore, displacement monitoring is not suitable for predicting block falls. In addition, the fall of a single block may lead to instantaneous instability of adjacent blocks. It is impossible to predict the collapsed space effectively by monitoring the displacement on the exposed surface of a single block. In conclusion, displacement prior to a block falling is not marked or predictable, making displacement monitoring unsuitable for predicting underground rockfall events.

### 3.2. Results of Temperature Information

During the test, the ambient temperature is approximately 15 °C. An infrared thermal imager is used to monitor the temperature response of unstable blocks and adjacent blocks during the collapse of dangerous rocks and reveal the response law of the rock mass temperature field during tunnel construction.

For direction ①, given block 4 and block 2 fell during the 11th and 21st steps, the steps from 10 to 12 and 20 to 22 were selected as intervals for studying the temperature response related to block fall. Three measuring lines, L1, L2, and L3, were selected from associated joints, and the variation of average temperature along each line was analyzed (Figure 10).

In the 10th step, the contact between blocks loosened, while average temperatures along L1, L2, and L3 all dropped by approximately 0.05 °C. In the 11th step, block 4 detached from block 2 and block 3, and the average temperature of L1 dropped sharply by 0.17 °C, while L2 and L3 showed little change. In the 12th step, no significant change in the average temperature was recorded along L1, L2, and L3, related to the lack of change along block contacts. Likewise, in the 20th step, average temperatures of L1, L2, and L3 all dropped approximately 0.05 °C. In the 21st step, block 2 detached from block 1 and block 3 and made contact with block 4. The average temperatures of L2 and L3 dropped sharply by 0.15 °C and 0.23 °C, respectively, while the average temperature of L1 rose by 0.06 °C. In the 22nd step, block 2 made contact with block 1 and block 3, and the average temperatures of L2 and L3 rose sharply by 0.26 °C and 0.24 °C, respectively, while the average temperature of L1 was unchanged.

For direction ②, as shown in Figure 11, in the 12th step, rock block 2 was completely exposed, the contact relationship between rock blocks 1, 3, and 4 and rock block 2 was loosened, and the temperature of structural surfaces L1, L2, and L3 dropped sharply by 0.05 °C. Afterward, the contact relationship of each rock block was adjusted. Due to the squeezing effect, the temperature of each structural surface gradually increased, until the 18th step of excavation was basically adjusted. At this time, the L1 structural surface had the highest temperature, and the temperature increased by approximately 0.16 °C. This shows that the structural plane L1 plays a major role in the stability of the structural plane. After the 19th step, rock block 2 was unstable, and the temperature of each structural surface suddenly dropped by approximately 0.15 °C. By the 20th step, rock block 2 was completely unstable, and the temperature field was relatively stable. 

Clearly, a rise or drop in temperature was accompanied by the detachment or contact of blocks. Considering that the essence of block fall requires detachment of the block from its parent rock, a temperature drop in related joints can be used as an indicator of imminent block fall. Typically, different rock block failure modes have different response laws. For falling-type instability rocks, the sudden drop in temperature of the controlled structural surface can be used as a precursor to instability. As for the slipping instability rock, there is the possibility of lagging instability. The temperature of the main control structure surface drops sharply and then rises gradually, which can be used as a precursor to the instability of the lagging slip instability rock.

### 3.3. Results of Natural Vibration Information

The test process uses the acceleration sensor to obtain the time-domain vibration information of the block, and then the time-domain information is transformed by FFT to obtain the frequency-domain information. The frequency value corresponding to the maximum amplitude value is the natural vibration frequency of the block at that moment. To study the variation of the natural vibration frequency during the rockfall process, we focused on block 4 and block 2, as shown in Figure 12 and Figure 13.

For direction ①, as shown in Figure 12, the variation of the natural vibration frequency of block 4 from initial exposure in the 1st step to complete exposure in the 11th step was studied in detail. The first four steps had little influence on the natural vibration frequency, resulting in a stable frequency of approximately 50 Hz. After the 4th step, the natural vibration frequency was reduced, decreasing to 45 Hz by the 6th step. Little change in natural vibration frequency occurred during the 7th and 8th steps, but it dropped to its lowest point in the 9th and 10th steps, reaching 42 Hz. Block 4 fell during the 11th step, after which its natural vibration frequency returned to the initial level of approximately 50 Hz. The variation of the natural vibration frequency of block 2 was also studied from initial exposure in the 12th step to complete exposure in the 21st step. The natural vibration frequency of block 2 decreased from 62 Hz to 45 Hz during the transition from the 12th step to the 18th step, respectively. After the 18th step, the upper left corner of block 2 made contact with block 4, and this contact area increased during ongoing excavation. During this stage, the natural vibration frequency of block 2 began to increase (19th step), reaching its highest level of 75 Hz during the 20th step. Block 2 fell during the 21st step, after which the natural vibration frequency decreased slightly to 72 Hz.

For direction ②, as shown in Figure 13, in the first step, the initial natural frequency of block 2 is around 50 Hz. With the increase in the excavation steps, the natural vibration frequency of block 2 has an overall downward trend. After the 17th step is completed, the natural vibration frequency is reduced to 32 Hz. When the 18th step of excavation was completed, the natural vibration frequency of block 2 increased slightly. Combined with the test results, it was found that due to the continuous excavation, the right foot of block 2 came into contact with the frame. After the 9th step of excavation, block 2 began to slide, and the natural vibration frequency began to decrease again. Until the 20th step, block 2 had completed the overall fall. At this time, blocks 1 and 3 were supported by the 21st and 22nd sets of jacks, and there is no obvious falling trend. After the 21st step, the contact surface of blocks 1, 3, and 4 and block 2 increased, causing the natural vibration frequency of block 2 to start to increase. Until the completion of the 22nd step, blocks 1, 3, and 4 all fell. At this time, the natural vibration frequency of block 2 returned to the initial higher level, with a frequency value of approximately 50 Hz.

Clearly, a rise or fall in natural vibration frequency was correlated with a decrease or increase in the exposed area of the block. Considering that block fall is often accompanied by an increase in exposure, the drop in the natural vibration frequency of a block could be used to indicate imminent block fall. The tunnel block fall disaster may be manifested as a chain block fall caused by the instability of a single block. The results of this test also show that the fall of a single block will produce a certain degree of the excitation effect on adjacent blocks, which is an induced lock potential influence factor of instability. Moreover, the drop in the natural vibration frequency of the block always occurred prior to block fall, suggesting it can be used to predict rockfalls.

### 3.4. Results of Acoustic Emission Information

The rock mass releases stored energy in the form of sound for a certain period before the rock mass collapses. The intensity of this energy release varies according to the proximity of the structural instability. In this experiment, an acoustic emission system is used to monitor the internal damage of rock mass during the rockfall, and the hit accumulation and energy accumulation of acoustic emission characteristics during the excavation process were studied (Figure 14).

For direction ①, as shown in Figure 14a, the hit accumulation and energy accumulation of acoustic emissions were characterized into three stages: a slow-growth stage, an accelerated-growth stage, and an instantaneous-burst stage, based on a detailed examination of the process of block 2 falling. During the first eight steps, the blocks were relatively stable, and the number of hits and energy accumulation were characterized by slow growth. From the 8th step to the 10th step, the influence of excavation on the stability of blocks increased, and the number of hits and energy accumulation were characterized by accelerated growth. In the 11th step, the contact relationships and the force between the blocks changed completely, and the number of hits and energy accumulation entered an Instantaneous-burst stage. The proportions of hits in the three stages were 7.05%, 11.78%, and 81.17%, respectively, while the proportions of energy accumulated were 1.44%, 3.09%, and 95.47%, respectively. The ratios of the proportions of energy accumulation to hit accumulation during the slow-growth, accelerated-growth, and instantaneous-burst stages were 0.20, 0.26, and 1.1, respectively. 

Following the fall of block 4, the original stable state of adjacent block 2 was undermined. Both hits and energy accumulation increased almost exponentially at this point, producing an accelerated-growth stage after the 11th step. This entered the instantaneous-burst stage in the 21st step. As block 2 made contact with block 4 in the 18th step, the energy release was no longer characterized by an instantaneous burst but rather was gradually released during excavation. The proportions of hits accumulated in these two stages were 91.84% and 8.16%, while the proportions of the energy accumulated were 86.87% and 13.33%, respectively. The ratios of the proportions of energy accumulation to hit accumulation during the accelerated-growth and instantaneous-burst stages were 0.95 and 1.63, respectively.

For direction ②, as shown in Figure 14b, the 1st to 10th excavation steps were relatively calm, and no acoustic emission phenomenon was detected. After the 11th excavation, the cumulative impact began to increase, and then the acoustic launch site was calm again with low activity. After the 18th excavation, many impacts were detected by the probe, and the cumulative impact curve showed an obvious increasing trend. At this time, combined with the test phenomenon, block 2 began to have sliding friction along the structural surface. After the excavation in the 20th step was completed, the sliding and falling of block 2 ended. During the excavation in the 21st and 22nd steps, blocks 1, 3, and 4 began to fall, and the cumulative impact increased further at this time. The proportions of hits in the three stages were 0.95%, 12.94%, and 62.7%, respectively, while the proportions of energy accumulated were 0.75%, 8.62%, and 59.16%, respectively. The ratios of the proportions of energy accumulation to hit accumulation during the slow-growth, accelerated-growth, and instantaneous-burst stages were 0.79, 0.67, and 0.94, respectively.

The evolution law of energy accumulation is like that of hit accumulation. Acoustic emission signals are received after the excavation in the 10th step, and the energy is relatively weak. During the 10th to 17th excavation steps, the energy accumulation curve fluctuates slightly, and the whole is in a quiet period. After the 18th step is completed, the acoustic emission energy value increases sharply. It is worth noting that the number of impacts during this step of the excavation process is greater than that of the 19th step, but the energy increase is almost the same for the two steps. After rock block 2 fell, with the 21st and 22nd steps of the excavation, more acoustic emission signals continued to be generated, and the cumulative energy curve increased significantly. Evidently, there is a surge in hits and energy accumulation before a block fall. The closer to instability the block is, the higher the energy release of a single hit. Therefore, the surge in hits and energy accumulation and the increase in single-hit energy could be regarded as effective precursor indicators of rockfall.

## 4. Discussion

There are currently two core research directions for rockfall monitoring in underground engineering. One is the real-time monitoring technology for dangerous rocks based on displacement and stress monitoring [22,23,24]. Traditional monitoring methods have a long history of development and have a good feedback effect on the deformation and stress of surrounding rock. However, compared with other disasters in underground engineering, rockfall has significant localization and strong suddenness. It is very difficult to realize effective monitoring of dangerous rocks under the condition that the starting point of instability is difficult to be accurately identified. With the development of full-range and all-round monitoring technologies such as laser scanners and laser radars, the whole process of monitoring the displacement is no longer a dream. Determining how to solve the real-time monitoring problem in the underground engineering construction environment is a key problem that needs to be solved urgently.

The second is the dangerous rock instability monitoring and early warning technology based on multi-physics field precursors [25,26,27,28,29,30]. The mechanical catastrophe of rocks and structural surfaces will produce a variety of physical effects, such as stress, temperature, deformation, acoustic emission, electromagnetic radiation, and other physical quantities of abnormal changes. These physical quantities are not isolated but rather interrelated and orderly. Through the correlation and combined analysis of various physical information, the reliability of rockfall identification can be improved. The problem that single information cannot identify or accurately identify the precursors can be solved. In this paper, by analyzing the evolution law of displacement, temperature, vibration information, and energy information during the rockfall process, it is found that the degree of construction excavation exposure has a significant impact on the natural vibration frequency of the block. The higher the degree of separation between the block and the parent rock, the lower the natural vibration frequency. Before the disaster occurs, the natural vibration frequency drops sharply, which can be used as a precursor to the rockfall disaster. At the same time, the cumulative impact number of acoustic emission signals and cumulative energy have the phenomenon of pro-disaster. Meanwhile, the prediction of the rockfall disaster range can be realized based on the location of the energy response. In summary, as shown in Table 2, the inherent vibration frequency and acoustic emission information can be used as effective monitoring indicators of rockfall in underground engineering.

## 5. Conclusions

In this paper, a large-scale three-dimensional experimental simulation apparatus to study failure mechanisms of rockfalls occurring during underground engineering is developed, and a complete simulation experiment of rockfalls in an underground engineering context was performed using this apparatus. Our conclusions are as follows:(1)A large-scale three-dimensional apparatus consisting of a prefabricated sub-system, a positioning sub-system, and a hydraulic sub-system was developed, with a size of 4 m × 4 m × 3.3 m. This experimental apparatus performed vertical (2.95 MPa) and horizontal symmetric loading (2.91 MPa) to accurately simulate the structure and stress environment of a rock mass.(2)The excavation process was simulated by releasing constraints of 22 basal hydraulic cylinders in a step-by-step manner, while displacement, temperature, natural vibration, and acoustic emissions during each step were monitored. The experimental apparatus simulated the entire process of a rockfall during underground excavation—including gestation, development, and formation stages. This allowed us to study the energy accumulation mechanism and dynamic evolution of rockfalls in underground engineering settings.(3)Complete simulation experiments of a rockfall were carried out, revealing that block fall occurs at the moment when the block is completely exposed. A temperature drop in related joints, a drop in the natural vibration frequency, and a surge in hits and energy accumulation could be used to predict rockfalls.

## Figures and Tables

**Figure 1 sensors-24-02068-f001:**
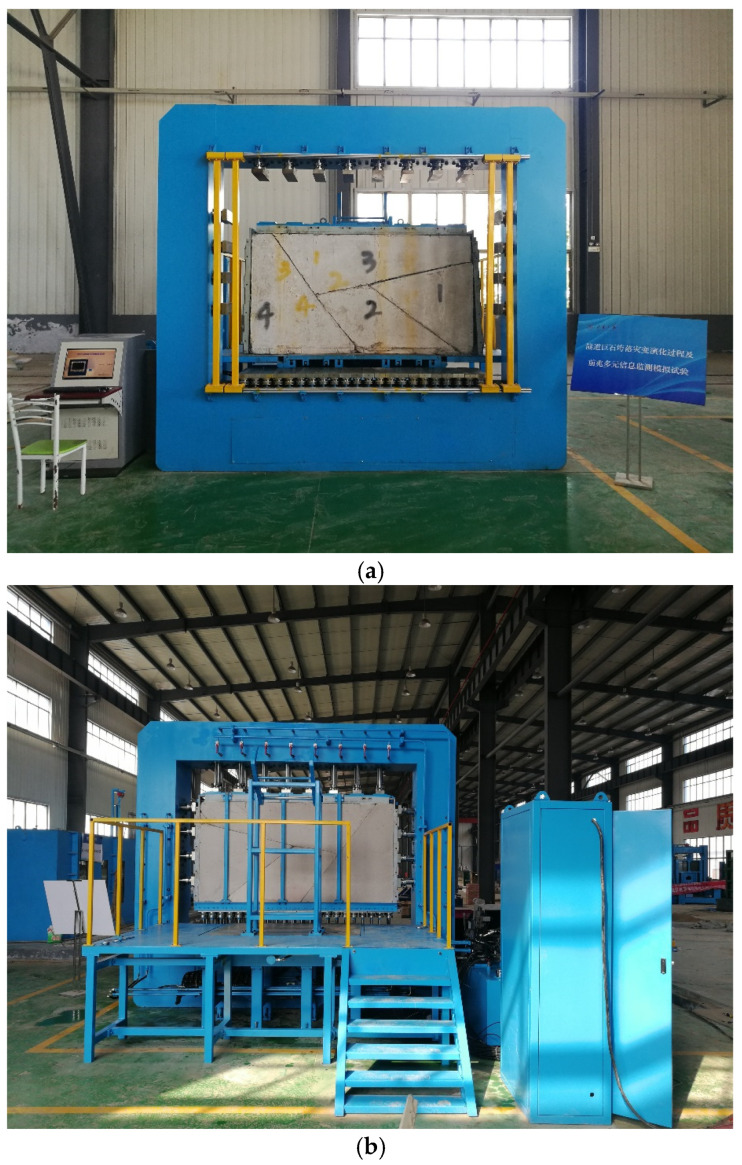
Large-scale three-dimensional rockfall simulation apparatus. (**a**) Front view; (**b**) rear view.

**Figure 2 sensors-24-02068-f002:**
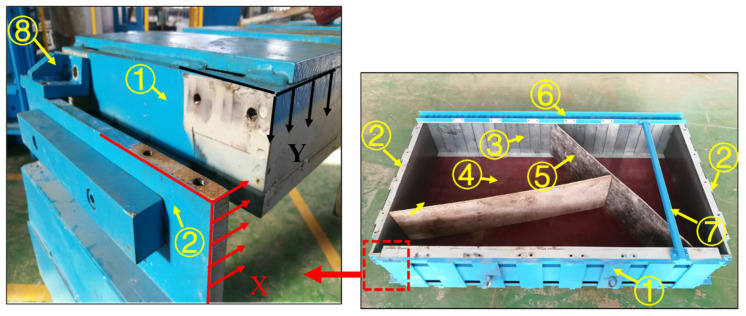
The structure of the prefabricated sub-system. ① Top loading plate. ② Lateral loading plate. ③ Bottom loading plate. ④ Bamboo veneers. ⑤ Prefabricated panel. ⑥ U-type connector. ⑦ Square connector. ⑧ L-type connector.

**Figure 3 sensors-24-02068-f003:**
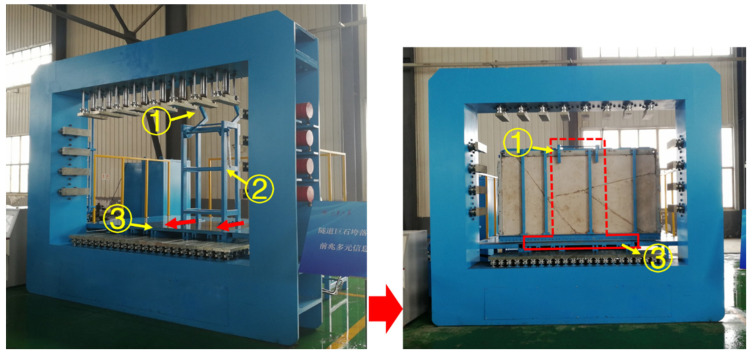
The structure of the positioning sub-system. ① The positioning hook. ② The balance bracket. ③ The platform with propulsion plant.

**Figure 4 sensors-24-02068-f004:**
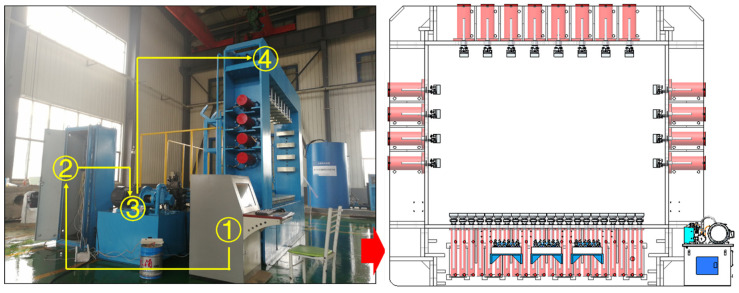
The structure of the hydraulic sub-system. ① The control platform. ② The main box. ③ The hydraulic press. ④ The hydraulic cylinders.

**Figure 5 sensors-24-02068-f005:**
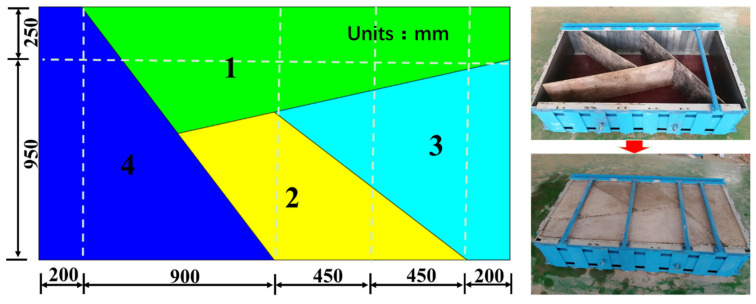
Structure and size of the model specimen.

**Figure 6 sensors-24-02068-f006:**
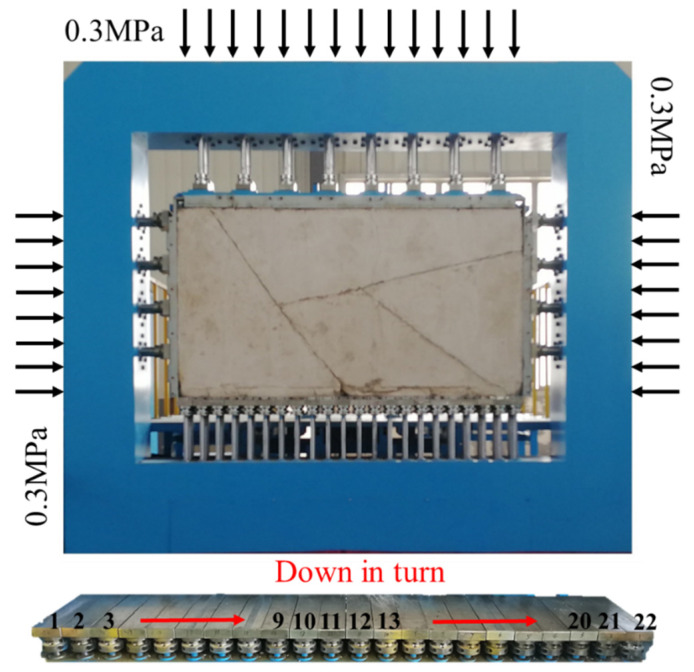
Boundary stress conditions during tunnel’s excavation simulation.

**Figure 7 sensors-24-02068-f007:**
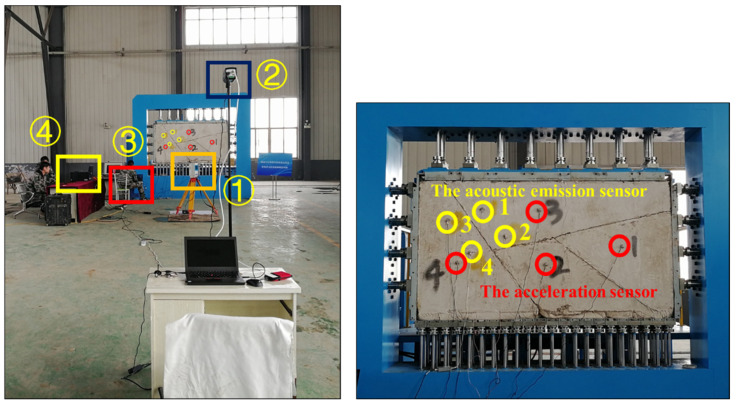
Monitoring instrument and monitoring design scheme. ① The laser scanner. ② The infrared thermal imager. ③ The acoustic emission system. ④ The vibration frequency acquisition instrument.

**Figure 8 sensors-24-02068-f008:**
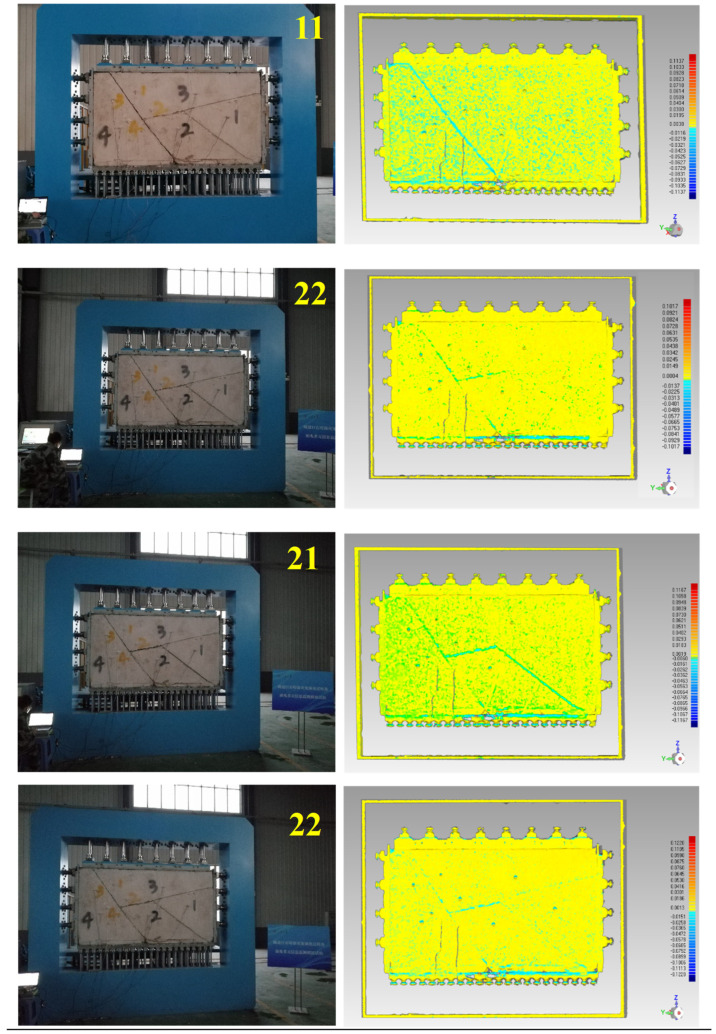
Results of displacement information during the excavation for direction ①.

**Figure 9 sensors-24-02068-f009:**
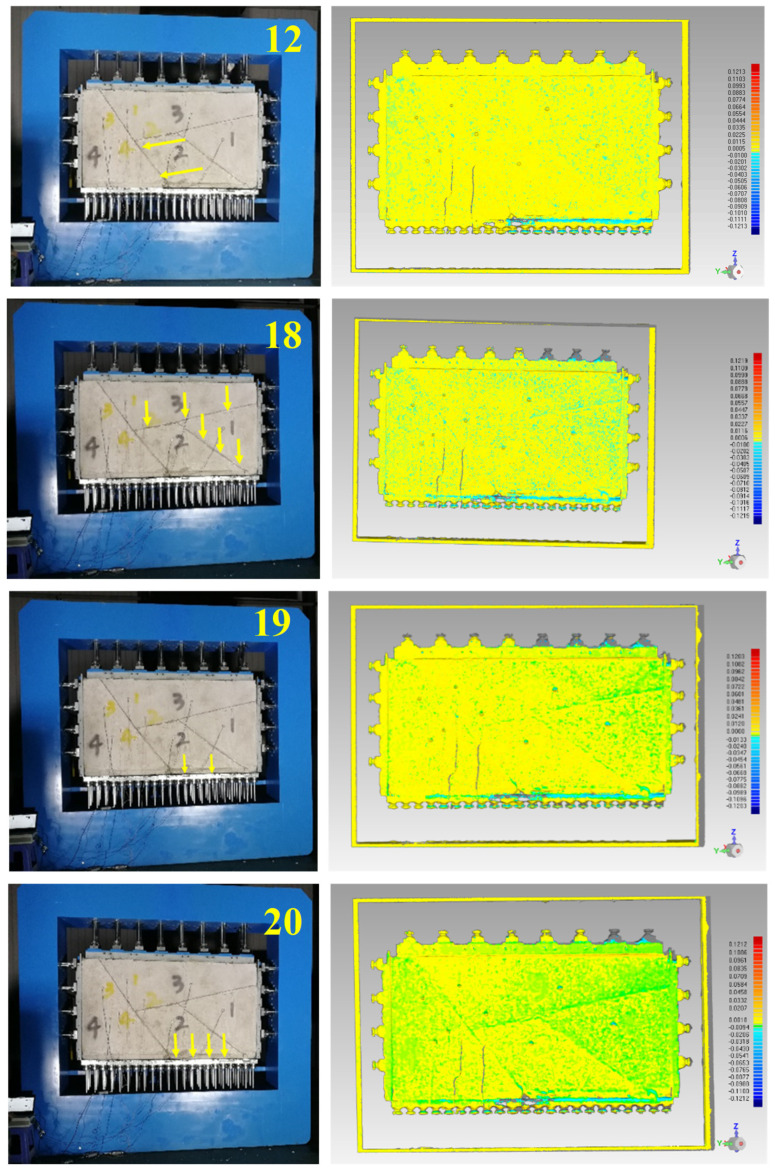
Results of displacement information during the excavation for direction ②.

**Figure 10 sensors-24-02068-f010:**
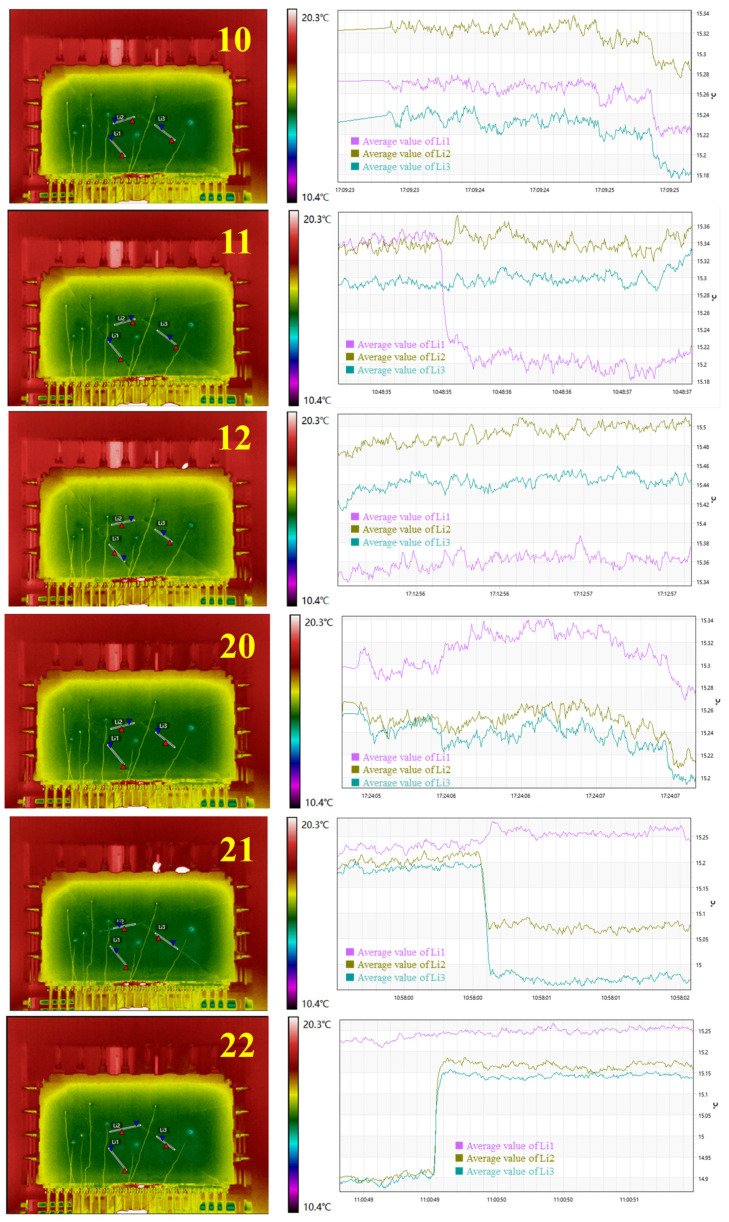
Results of temperature information during the excavation for direction ①.

**Figure 11 sensors-24-02068-f011:**
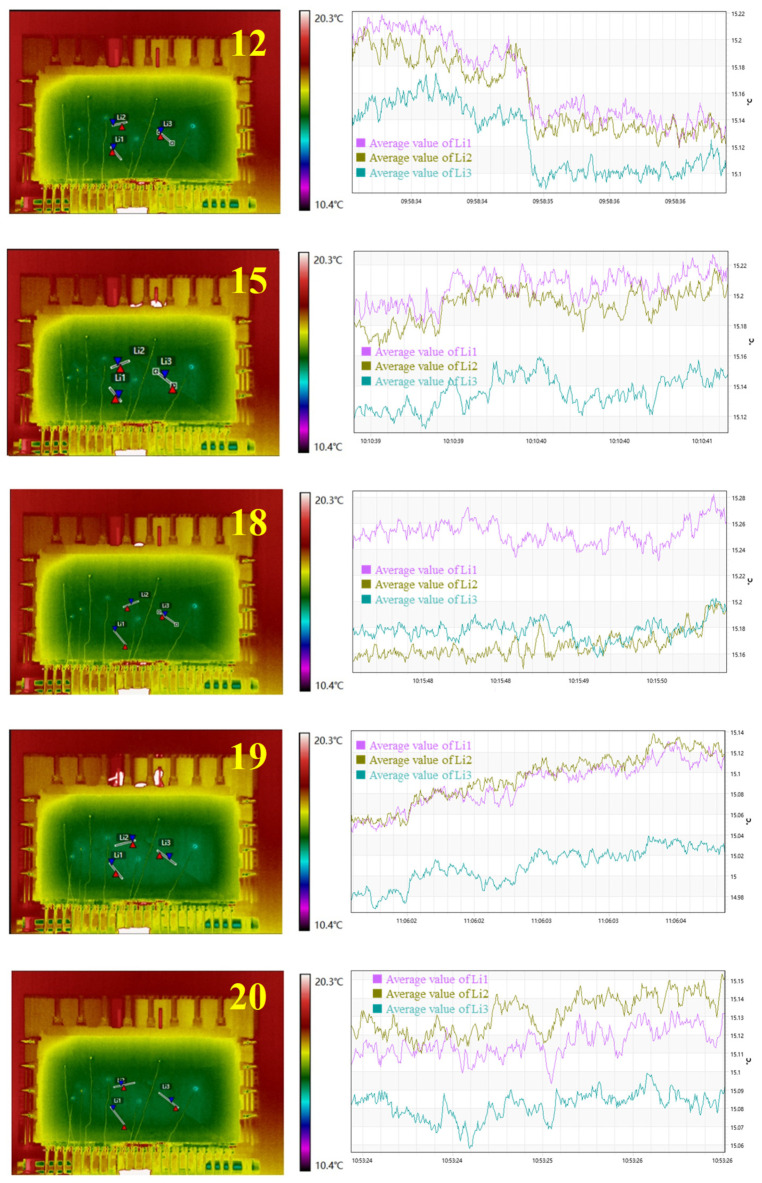
Results of temperature information during the excavation for direction ②.

**Figure 12 sensors-24-02068-f012:**
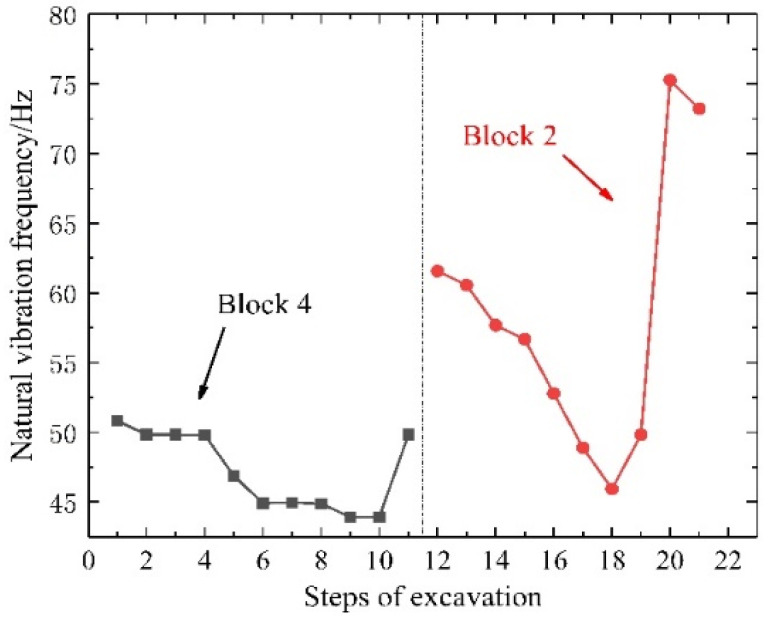
Natural vibration frequency of each step during the excavation for direction ①.

**Figure 13 sensors-24-02068-f013:**
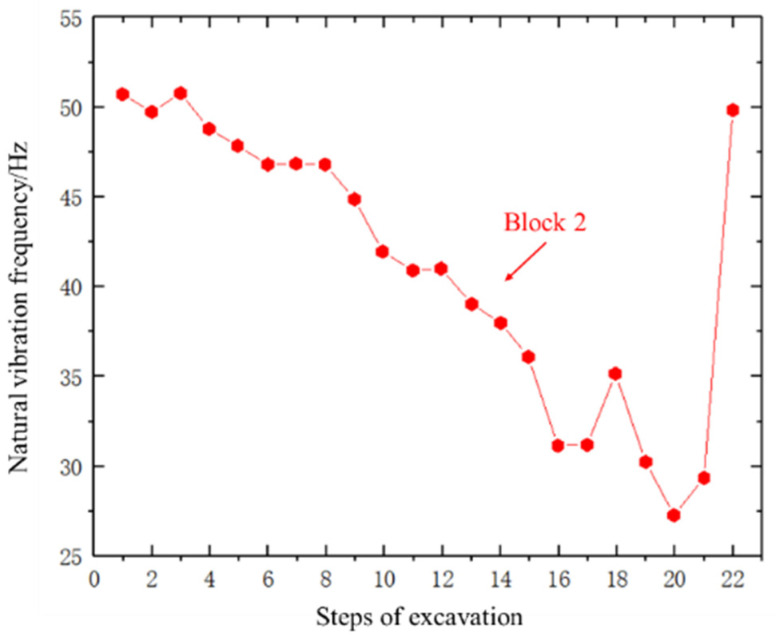
Natural vibration frequency of each step during the excavation for direction ②.

**Figure 14 sensors-24-02068-f014:**
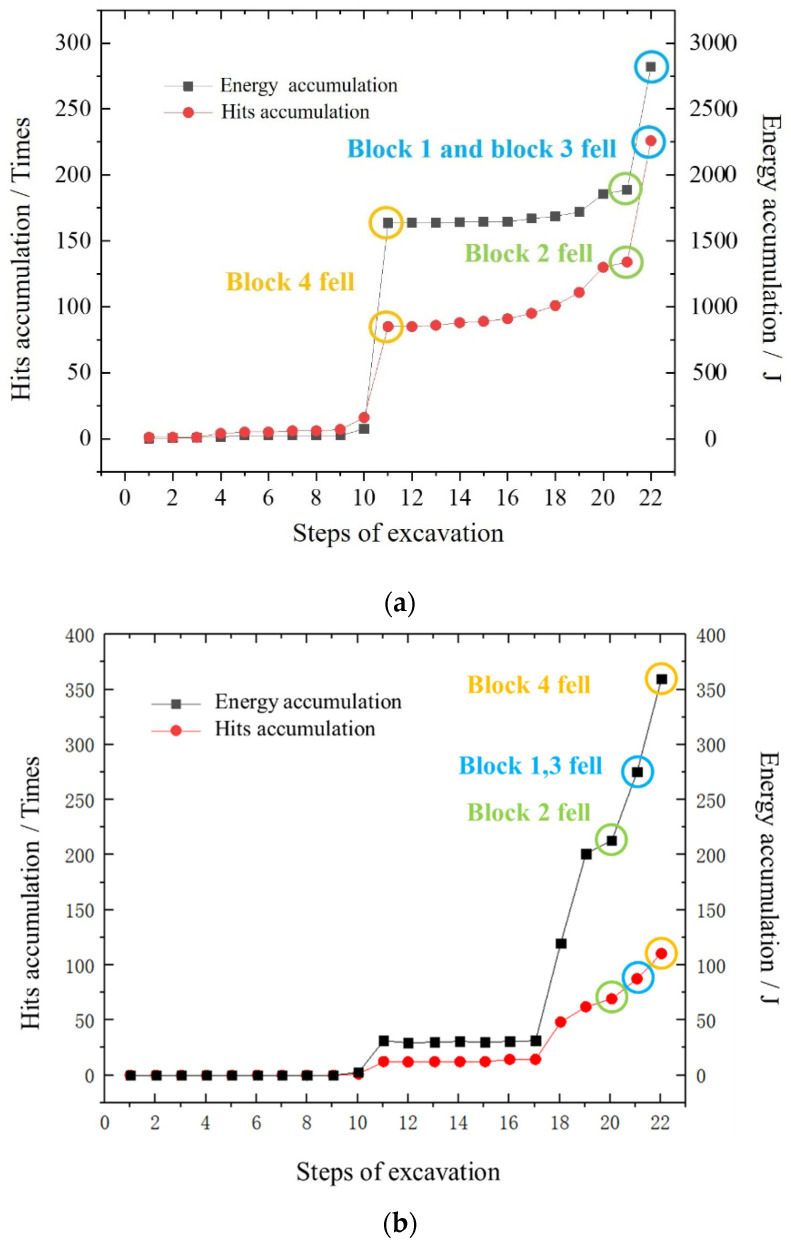
Acoustic emission characteristics of each step during the excavation. (**a**) Acoustic emission characteristics of each step for direction ①; (**b**) Acoustic emission characteristics of each step for direction ②.

**Table 1 sensors-24-02068-t001:** Main parameters of the experimental apparatus.

Size of Apparatus	4 m × 4 m × 3.3 m	Size of Model Specimen	2.2 m × 0.4 m × 1.2 m
Size of top loading plate	2.2 m × 0.4 m × 0.115 m	Size of lateral loading plate	1.2 m × 0.4 m × 0.65 m
Size of bottom loading plate	0.1 m × 0.4 m × 0.02 m	Stroke of top hydraulic cylinder	0.2 m
Stroke of lateral hydraulic cylinder	0.2 m	Stroke of bottom hydraulic cylinder	0.4 m
Vertical loading pressure	2600 kN (2.95 MPa)	Horizontal loading pressure	1400 kN (2.91 MPa)
Continuous loading time	≥720 h	Pressure control accuracy	±6‰

**Table 2 sensors-24-02068-t002:** Rockfall physical information sensitivity ranking.

Disaster Type	Falling Type	Slipping Type
Temperature	The response is not obvious	The response is not obvious
Displacement	The response is not obvious	Increase with excavation
Natural vibration frequency	Drops sharply	Drops sharply
Acoustic emission (microseismic)	Increase sharply	Increase sharply

## Data Availability

All data, models, and code that support the findings of this study are available from the corresponding author upon reasonable request.

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
