# Peer review of "A Large-Scale Three-Dimensional Apparatus to Study Failure Mechanisms of Rockfalls in Underground Engineering Contexts"

_sensors, 2024, doi:10.3390/s24072068_

Round 1
Reviewer 1 Report
Comments and Suggestions for Authors
- The work suffers from superficial literature review. Without any doubt MUST be decorated with more relevant and updated works.
- The lack of innovation and highlighted research gaps must be fulfilled.
- Obvious available inconsistencies in the reference list MUST be treated.
- This paper doesn’t have real Discussion because of inadequate accuracy performance metrics, evidential analysis and physical interpretation, solid fair comparison with other scholars, limitation of this work, stability approval of the results, pitfalls and practical difficulties of applied model under certainty, impact of the bias of the used data on the results…)… overall, WONDERED with Discussion without any citations!!!!
- Any Discussion concerning the rock type, rock mass, disconformities, …
- Lack of considering the spatial subsurface geomaterial distributions.
- Couldn’t follow the experiment and simulate the real conditions.
- How the spatial distributions in Figs 8, and 9 were achieved. What can the used legend depict??
- Plots for Fig 12 are unreadable and in Chinese!!!
- Concerning the Fig 12 and time dependent records, 1) how did you treat the heterogeneity of data as well as outliers?? The unhandled heterogeneities in the records obviously indicate that the results mathematically cannot provide any generalization from a single study. Therefore, the given results analytically suffer from accurately identifying the correct model to represent the data. Are all the parameters measured at the same time??
- The inclusion of uncertainty in this experimental work is very high. How did you get involved in the analysis?? Clarify how the uncertainty will be quantified and with what criteria you can provide reliable interpretation. Maybe beneficial to have a look at https://link.springer.com/article/10.1007/s00366-023-01852-5... (The concept of the work is subject of matter)
- Considering Fig 12 is another story which should be clarified through PPV. Considering the frequency requires long discussion and interpretation which this work suffers from.
- Unjustified conclusion.
Comments on the Quality of English Language
- English of the work should definitely be proofread by a native expert.
Author Response
Dear reviewers:
Thank you for your letter and the reviewers’ comments on our manuscript entitled “A large-scale three-dimensional apparatus to study failure mechanisms of rockfalls in underground engineering contexts”. Those comments are very helpful for revising and improving our paper, as well as the important guiding significance to other research. We have studied the comments carefully and made corrections which we hope meet with approval. The main corrections are in the manuscript and the responds to the reviewers’ comments are as follows (the replies are highlighted in blue).
Replies to the reviewers’ comments:
Reviewer 1
(1) The work suffers from superficial literature review. Without any doubt MUST be decorated with more relevant and updated works.
The lack of innovation and highlighted research gaps must be fulfilled.
According to the comments of the reviewers, we have rewritten the Introduction section to provide a detailed supplement to the relevant background of the study, the relevant contents are highlighted in blue.
(2) Obvious available inconsistencies in the reference list MUST be treated.
According to the comments of the reviewers, we have treated the Reference.
(3) This paper doesn’t have real Discussion because of inadequate accuracy performance metrics, evidential analysis and physical interpretation, solid fair comparison with other scholars, limitation of this work, stability approval of the results, pitfalls and practical difficulties of applied model under certainty, impact of the bias of the used data on the results…)… overall, WONDERED with Discussion without any citations!!!!
Any Discussion concerning the rock type, rock mass, disconformities, …
(4) Lack of considering the spatial subsurface geomaterial distributions.
This research is mainly applicable to fractured rock mass, rather than soft rock, etc.
(5) Couldn’t follow the experiment and simulate the real conditions.
The test structure simulates the axial direction of the tunnel, rather than the cross-section of the tunnel. Maybe this can make people understand the condition simulated by the test better.
(6) How the spatial distributions in Figs 8, and 9 were achieved. What can the used legend depict??
Figs 8, and 9 are the displacement monitoring result of the three-dimensional laser scanner. In this test, a Z+F 5010C laser scanner is used to monitor the displacement field of the whole blocky rock mass in real time
(9) Plots for Fig 12 are unreadable and in Chinese!!!
According to the comments of the reviewers, we have re-edited Fig 12 and 13.
(10) Concerning the Fig 12 and time dependent records, 1) how did you treat the heterogeneity of data as well as outliers?? The unhandled heterogeneities in the records obviously indicate that the results mathematically cannot provide any generalization from a single study. Therefore, the given results analytically suffer from accurately identifying the correct model to represent the data. Are all the parameters measured at the same time??
The inclusion of uncertainty in this experimental work is very high. How did you get involved in the analysis?? Clarify how the uncertainty will be quantified and with what criteria you can provide reliable interpretation. Maybe beneficial to have a look at https://link.springer.com/article/10.1007/s00366-023-01852-5... (The concept of the work is subject of matter)
Considering Fig 12 is another story which should be clarified through PPV. Considering the frequency requires long discussion and interpretation which this work suffers from.
The data in Figure 12 and Figure 13 are the natural vibration frequencies that have been processed.
The test process uses the acceleration sensor to obtain the time domain vibration information of the block, and then the time-domain information was transformed by FFT to obtain the frequency-domain information. The frequency value corresponding to the maximum amplitude value is the natural vibration frequency of the block at that moment.
(13) Unjustified conclusion.
According to the comments of the reviewers, we rewrote the Conclusion section, the relevant contents are highlighted in blue.
Reviewer 2 Report
Comments and Suggestions for Authors
Review of the article
A large-scale three-dimensional apparatus to study failure
mechanisms of rockfalls in underground engineering contexts
The article is devoted to the development and testing of an experimental setup (apparatus) for identification of rock structure. In our opinion, the article has some practical sense in terms of determining the load that causes disturbances in the rock structure.
However, there are such questions and comments:
1. Disturbance of rock structure, such as cracks and loss of adhesion between elements is not always a condition for loss of bearing capacity and rockfall of the rock, because separated elements can be wedged, held by adhesion, irregularities, etc.
In this sense, full-scale testing of the rock is more objective.
Please clarify this point, is it possible to determine such forces on the proposed installation?
2. The components of the apparatus are poorly described. In particular, such an important subsystem as hydraulics should have an appropriate hydraulic diagram with indication of pumps, throttles, check valves and other elements. The same applies to the temperature and vibration subsystems.
3. Many of the diagrams do not show numbers and letters, they are too small.
4. The conclusions of the article should be reworded. In particular, what is the scientific result in conclusion (1). The new device is not a scientific novelty but an engineering result. In Conclusion (3), in my opinion, it should be formulated as a disruption of the rock structure, but not a prediction of rockfall. Why? see point 1, and the stability of the rock massif and rockfall cannot be objectively predicted in the apparatus, since in reality there may be a number of factors influencing this process.
The article may be published after revision.
Sincerely, reviewer
Author Response
Dear reviewers:
Thank you for your letter and the reviewers’ comments on our manuscript entitled “A large-scale three-dimensional apparatus to study failure mechanisms of rockfalls in underground engineering contexts”. Those comments are very helpful for revising and improving our paper, as well as the important guiding significance to other research. We have studied the comments carefully and made corrections which we hope meet with approval. The main corrections are in the manuscript and the responds to the reviewers’ comments are as follows (the replies are highlighted in blue).
Replies to the reviewers’ comments:
Reviewer 2
- Disturbance of rock structure, such as cracks and loss of adhesion between elements is not always a condition for loss of bearing capacity and rockfall of the rock, because separated elements can be wedged, held by adhesion, irregularities, etc.
In this sense, full-scale testing of the rock is more objective.
Please clarify this point, is it possible to determine such forces on the proposed installation?
Our research mainly focuses on disasters such as falling blocks, and the research is based on the block theory proposed by Professor Shi Genhua. The main characteristic of this kind of damage is that the main control structural plane is damaged, but the rock itself is not damaged. Therefore, many of our studies are mainly focused on the main control structural plane.
The wedged, held by adhesion, irregularities, etc. you proposed can be simulated during the rock mass prefabrication process.
And we're not quite sure what you mean by comprehensive testing of the rocks.
- The components of the apparatus are poorly described. In particular, such an important subsystem as hydraulics should have an appropriate hydraulic diagram with indication of pumps, throttles, check valves and other elements. The same applies to the temperature and vibration subsystems.
According to the comments of the reviewers, we have supplied the description in fig.4, the relevant contents are highlighted in blue.
- Many of the diagrams do not show numbers and letters, they are too small.
According to the comments of the reviewers, we have re-edited all the diagrams.
- The conclusions of the article should be reworded. In particular, what is the scientific result in conclusion (1). The new device is not a scientific novelty but an engineering result. In Conclusion (3), in my opinion, it should be formulated as a disruption of the rock structure, but not a prediction of rockfall. Why? see point 1, and the stability of the rock massif and rockfall cannot be objectively predicted in the apparatus, since in reality there may be a number of factors influencing this process.
According to the comments of the reviewers, we rewrote the Conclusion section, the relevant contents are highlighted in blue.
Round 2
Reviewer 1 Report
Comments and Suggestions for Authors
Thanks for the responses, but such responding is DECLINED and cannot be followed:
1. WONDERED why several comments didn’t have been responded??? Why their ordinal number is not consistent??? They MUST be replied to one-by-one for the sub-comments.
For example: #3, , ‘Any Discussion…’, #10, ‘The inclusion of uncertainty …’, ‘Considering Fig 12…’…
2. Seems that some of the comments are eliminated!!!! for example, the need for English proofread, #7, #8, …
3. Some of the comments are totally misunderstood, #4, #5, …
4. The template has the continues line numbers which can be used to assign the exact place of the modifications. Assign the corresponding line number for each comment to facilitate tracking. This is not the reviewers ‘ task to look up the context and find your response.
5. The responses are not just for the reviewer but also maybe the question of future readers. Therefore, adding extra descriptive statements for more clarification is mandatory.
Comments on the Quality of English Language
English must be gone under professional proofread by a native agent
Author Response
Dear reviewers:
Thank you for your letter and the reviewers’ comments on our manuscript entitled “A large-scale three-dimensional apparatus to study failure mechanisms of rockfalls in underground engineering contexts”. Those comments are very helpful for revising and improving our paper, as well as the important guiding significance to other research. We have studied the comments carefully and made corrections which we hope meet with approval. The main corrections are in the manuscript and the responds to the reviewers’ comments are as follows (the replies are highlighted in blue).
Replies to the reviewers’ comments:
Reviewer 1
- WONDERED why several comments didn’t have been responded??? Why their ordinal number is not consistent??? They MUST be replied to one-by-one for the sub-comments.
For example: #3, , ‘Any Discussion…’, #10, ‘The inclusion of uncertainty …’, ‘Considering Fig 12…’…
According to the comments of the reviewers, we have responded to the first comments one by one. The following is the new reply:
(1) The work suffers from superficial literature review. Without any doubt MUST be decorated with more relevant and updated works.
According to the comments of the reviewers, we have rewritten the Introduction section to provide a detailed supplement to the relevant background of the study, the relevant contents are highlighted in blue in line 27-92.
(2) The lack of innovation and highlighted research gaps must be fulfilled.
According to the comments of the reviewers, we have rewritten the Introduction section to provide a detailed supplement to the relevant background of the study, the relevant contents are highlighted in blue in line 27-92.
(3) Obvious available inconsistencies in the reference list MUST be treated.
According to the comments of the reviewers, we have treated the Reference in line 508-576.
(4) This paper doesn’t have real Discussion because of inadequate accuracy performance metrics, evidential analysis and physical interpretation, solid fair comparison with other scholars, limitation of this work, stability approval of the results, pitfalls and practical difficulties of applied model under certainty, impact of the bias of the used data on the results…)… overall, WONDERED with Discussion without any citations!!!!
According to the comments of the reviewers, we have rewritten the Discussion section to provide a more in-depth discussion of the research content of this article, the relevant contents are highlighted in blue in line 448-480.
(5) Any Discussion concerning the rock type, rock mass, disconformities, …
According to the comments of the reviewers, we have rewritten the Discussion section to provide a more in-depth discussion of the research content of this article, the relevant contents are highlighted in blue in line 448-480.
(6) Lack of considering the spatial subsurface geomaterial distributions.
This research is mainly applicable to fractured rock mass, rather than soft rock, etc.
(7) Couldn’t follow the experiment and simulate the real conditions.
The test structure simulates the axial direction of the tunnel, rather than the cross-section of the tunnel. Maybe this can make people understand the condition simulated by the test better.
(8) How the spatial distributions in Figs 8, and 9 were achieved. What can the used legend depict??
Figs 8, and 9 are the displacement monitoring result of the three-dimensional laser scanner. In this test, a Z+F 5010C laser scanner is used to monitor the displacement field of the whole blocky rock mass in real time
(9) Plots for Fig 12 are unreadable and in Chinese!!!
According to the comments of the reviewers, we have re-edited Fig 12 and 13.
(10) Concerning the Fig 12 and time dependent records, 1) how did you treat the heterogeneity of data as well as outliers?? The unhandled heterogeneities in the records obviously indicate that the results mathematically cannot provide any generalization from a single study. Therefore, the given results analytically suffer from accurately identifying the correct model to represent the data. Are all the parameters measured at the same time??
The data in Figure 12 and Figure 13 are the natural vibration frequencies that have been processed.
The test process uses the acceleration sensor to obtain the time domain vibration information of the block, and then the time-domain information was transformed by FFT to obtain the frequency-domain information. The frequency value corresponding to the maximum amplitude value is the natural vibration frequency of the block at that moment.
(11) The inclusion of uncertainty in this experimental work is very high. How did you get involved in the analysis?? Clarify how the uncertainty will be quantified and with what criteria you can provide reliable interpretation. Maybe beneficial to have a look at https://link.springer.com/article/10.1007/s00366-023-01852-5... (The concept of the work is subject of matter)
The data in Figure 12 and Figure 13 are the natural vibration frequencies that have been processed.
The test process uses the acceleration sensor to obtain the time domain vibration information of the block, and then the time-domain information was transformed by FFT to obtain the frequency-domain information. The frequency value corresponding to the maximum amplitude value is the natural vibration frequency of the block at that moment.
(12) Considering Fig 12 is another story which should be clarified through PPV. Considering the frequency requires long discussion and interpretation which this work suffers from.
According to the comments of the reviewers, we have provided long discussion and interpretation, the relevant contents are highlighted in blue in line 342-384.
(13) Unjustified conclusion.
According to the comments of the reviewers, we rewrote the Conclusion section, the relevant contents are highlighted in blue in line 483-502.
- Seems that some of the comments are eliminated!!!! for example, the need for English proofread, #7, #8, …
According to the comments of the reviewers, we have responded to the first comments one by one.
- Some of the comments are totally misunderstood, #4, #5, …
According to the comments of the reviewers, we have responded to the first comments one by one.
- The template has the continues line numbers which can be used to assign the exact place of the modifications. Assign the corresponding line number for each comment to facilitate tracking. This is not the reviewers ‘ task to look up the context and find your response.
According to the comments of the reviewers, we have assigned the corresponding line number for each comment to facilitate tracking.
- The responses are not just for the reviewer but also maybe the question of future readers. Therefore, adding extra descriptive statements for more clarification is mandatory.
According to the comments of the reviewers, we have added extra descriptive statements for more clarification is mandatory.

Reviewer 2 Report
Comments and Suggestions for Authors
Dear authors, your revisions are generally satisfactory.
Author Response
Dear reviewers:
Thank you for your letter and the reviewers’ comments on our manuscript entitled “A large-scale three-dimensional apparatus to study failure mechanisms of rockfalls in underground engineering contexts”. Those comments are very helpful for revising and improving our paper, as well as the important guiding significance to other research. We have studied the comments carefully and made corrections which we hope meet with approval.
Round 3
Reviewer 1 Report
Comments and Suggestions for Authors
Not getting appropriate feedback. One simple reason, you talk about the natural vibration frequency, which is processed, it doesn’t have any involved uncertainty??
Furthermore, citations within the context based on template should be embedded within the [..] not superscripts.
Are the all references new (as you make them tracked change)???
Comments on the Quality of English LanguagePlease double check the English as previously emphasized.
Author Response
Dear reviewers:
Thank you for your letter and the reviewers’ comments on our manuscript entitled “A large-scale three-dimensional apparatus to study failure mechanisms of rockfalls in underground engineering contexts”. Those comments are very helpful for revising and improving our paper, as well as the important guiding significance to other research. We have studied the comments carefully and made corrections which we hope meet with approval. The main corrections are in the manuscript and the responds to the reviewers’ comments are as follows (the replies are highlighted in blue).
Replies to the reviewers’ comments:
Reviewer 1
- Not getting appropriate feedback. One simple reason, you talk about the natural vibration frequency, which is processed, it doesn’t have any involved uncertainty??
The natural vibration frequency is the inherent property of the surrounding rock of the object. Surrounding rocks, blocks, and main structural surfaces can be considered as dynamic systems composed of physical parameters such as stiffness, mass, and damping. The stable surrounding rock around the block can be regarded as an infinite base relative to the block. The overall stiffness of the surrounding rock and block is relatively large, while the stiffness of the main structural surface is significantly weaker than that of the surrounding rock and block. When the main control structural surface of the block is damaged, the result will cause changes in the physical properties of the system, resulting in changes in the natural vibration frequency.
Therefore, it can be considered that when the mass of the block remains unchanged, the change in the natural vibration frequency of the block can reflect the degree of loss of the main control structural surface, thereby reflecting the stability of the block.
- Furthermore, citations within the context based on template should be embedded within the [..] not superscripts.
According to the comments of the reviewers, we have modified the citations within the context embedded within the [..], the relevant contents are highlighted in blue.
- Are the all references new (as you make them tracked change)???
According to the comments of the reviewers, we have added some new references and removed some old ones.